# Occult Breast Cancer Presenting as Sternum Pain

**DOI:** 10.3390/diagnostics14070765

**Published:** 2024-04-04

**Authors:** Dang Wu, Siyu Guo, Bicheng Zhang, Fengbo Huang, Wei Qian, Fuming Qiu, Qichun Wei, Ting Zhang

**Affiliations:** 1Department of Radiation Oncology, Second Affiliated Hospital, Zhejiang University School of Medicine, Zhejiang University, Hangzhou 310009, China; wudang@zju.edu.cn (D.W.); 22218721@zju.edu.cn (S.G.); bicheng_zhang@zju.edu.cn (B.Z.); qichun_wei@zju.edu.cn (Q.W.); 2Cancer Institute (Key Laboratory of Cancer Prevention and Intervention, National Ministry of Education), Second Affiliated Hospital, Zhejiang University School of Medicine, Zhejiang University, Hangzhou 310009, China; 2515183@zju.edu.cn (F.H.); qiufuming@zju.edu.cn (F.Q.); 3Department of Pathology, Second Affiliated Hospital, Zhejiang University School of Medicine, Zhejiang University, Hangzhou 310009, China; 4Department of Radiology, Second Affiliated Hospital, Zhejiang University School of Medicine, Zhejiang University, Hangzhou 310009, China; 2514010@zju.edu.cn; 5Department of Oncology, Second Affiliated Hospital, Zhejiang University School of Medicine, Zhejiang University, Hangzhou 310009, China

**Keywords:** sternum pain, occult breast cancer, PET-CT, IHC

## Abstract

Bone metastasis has been reported in up to 70% of patients with advanced breast cancer. A total of 55.76% of skeletal metastases in women were derived from breast cancer. However, patients with bone metastasis from an occult primary breast cancer are a rare subset of patients. Here, we present the case of a 38-year-old woman who had sternum pain for 4 months. A whole-body PET-CT scan revealed that the FDG uptake of both the sternum and internal mammary node was significantly increased. The final diagnosis of occult breast cancer was established by immunohistochemical (IHC) staining, which is of great significance for identifying the origin of a metastatic tumor despite no visualized lesions of mammary glands.

Occult breast cancer (OBC) is defined as an undetectable primary breast tumor presenting with clinically recognizable metastatic cancer [1]. It accounts for 0.3–1% of all breast cancers [2], and axillary and cervical lymph node metastasis is the common first clinical presentation [3]. Bone metastasis has been reported in up to 70% of patients with advanced breast cancer [4]. However, patients with bone metastasis from an occult primary breast cancer are a rare subset of patients, especially those with sternum metastasis [5]. Here, we present the case of a 38-year-old woman who had sternum pain for 4 months.

The patient also had an occasional cough. Physical examination revealed normal vital signs and tenderness in the sternum. A plain CT of her chest showed bone destruction at the level of sternal notches 2–4, with soft tissue mass (Figure 1A,B). A subsequent whole-body PET-CT scan revealed that the FDG uptake of the sternum and internal mammary node was significantly increased and other lesions with abnormal FDG uptake were not found (Figure 1C,D). To rule out multiple myeloma, serum-free light chain analysis displayed that serum-free light chain kappa was slightly increased (concentration 23.3 mg/L). Urine protein electrophoresis demonstrated a slight increase in urine protein (0.16 g/24 h). The other results of serum and urine protein electrophoresis and immunofixation, serum β2 microglobulin and serum concentration of common tumor markers were normal. In order to further determine the characteristics of her sternum lesions, sternum biopsy was performed. The histopathology showed that the tumor cells were infiltrated in the form of nests and cords in the bone tissue and skeletal muscle.

IHC staining showed that tumor cells were diffuse positive for cytokeratin and breast-specific markers GATA3 and GCDFP-15. The tumor cells expressed estrogen receptor (ER) at 60% (Figure 2B), progesterone receptor (PR) at 5% (Figure 2C), and androgen receptor (AR) at 80%, a Her-2/neu score of 1+ (Figure 2E) and negative for CK5/6 (Figure 2F). The proliferation index of Ki-67 was up to 80% (Figure 2D). However, the tumor cells were negative for Vimintin, TTF1, PAX8, P40, and CgA. Therefore, the histopathological and IHC evaluation of sternum lesions was consistent with metastatic breast carcinoma, luminal type B, despite no significant abnormality being observed in the breast MRI (Figure 3).

After diagnosis, the patient received systemic therapy with endocrine therapy and a CDK4/6 inhibitor, and local radiotherapy (RT) to sternum and internal mammary node lesions (Figure 4A,B). After local RT, a CT scan was performed to evaluate their therapeutic effect. The sternum and internal mammary node lesions were significantly receded after RT (Figure 5A,B).

Recently, ER+/HER2+ patients have been reported as more likely to have metastases to bone compared to ER−/HER2+ patients in a nationwide autopsy study using artificial intelligence [6]. In this case, the patient was diagnosed as luminal type B of occult breast carcinoma. According to the above study, the molecular type of this patient has a higher possibility of bone metastasis, which might support the finding of the sternum as the first metastatic site. However, another recent study demonstrated that the majority of distant metastases in advanced breast cancer putatively arise from occult regional nodal disease [7]. An interesting feature in this case was that the internal mammary node was also involved in soft tissue mass around sternum lesions, while the FDG uptake of axillary and supraclavicular lymph nodes, with a higher possibility of metastasis, was normal. Therefore, it was difficult to determine whether the sternum was the first metastatic site to invade the internal mammary lymph nodes or vice versa.

Occult breast cancer presenting as sternum pain is an uncommon condition. Any non-specific symptoms, such as bone pain as in the present case, should require high vigilance for occult breast cancer. The role of whole-body PET-CT in assessing possible systemic involvement and the potential primary tumor site is important. IHC is of great significance to identify the origin of a metastatic tumor despite no positive signs in mammary glands.

## Figures and Tables

**Figure 1 diagnostics-14-00765-f001:**
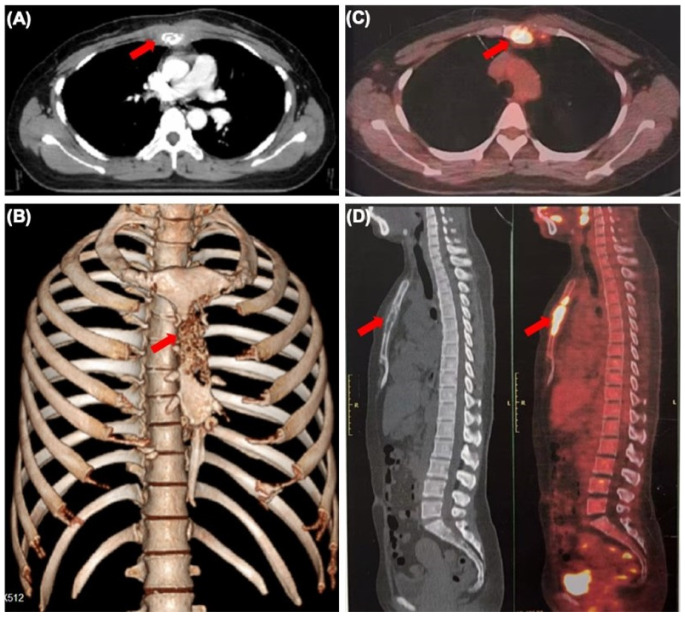
Sternum metastasis with soft tissue mass from occult breast cancer. (**A**) CT scan showing lytic lesions in the sternum (arrow). (**B**) Three-dimensional reconstruction of CT scan showing sternum lesion (arrow). (**C**,**D**) The whole-body PET-CT scan showing increased sternum FDG uptake (arrow).

**Figure 2 diagnostics-14-00765-f002:**
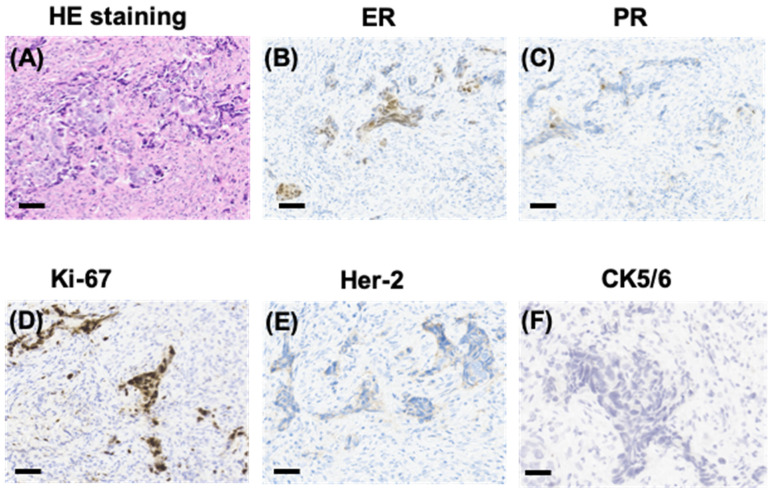
Pathological manifestations of the sternum lesions. HE staining showing sternum lesions (**A**) and immunohistochemical (IHC) staining showing positivity for ER (**B**), PR (**C**), and Ki-67 (**D**), and a Her-2/neu score of 1+ (**E**), and negativity for CK5/6 (**F**) in sternum tissue (scale bars represent 200 µm).

**Figure 3 diagnostics-14-00765-f003:**
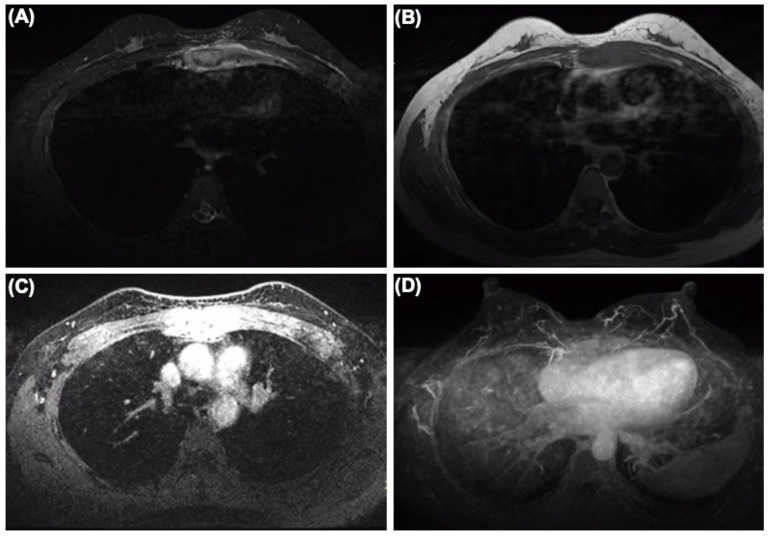
The MRI of the breast showing sternum and internal mammary node lesions without visualized lesions of both mammary glands and bilateral axillary lymph nodes. (**A**) SPIR (Spectral Presaturaton with Inversion Recovery) T2-weighted MR image. (**B**) T1-weighted MR image. (**C**) SPIR T1-weighted MR image. (**D**) Maximum density projection reconstruction.

**Figure 4 diagnostics-14-00765-f004:**
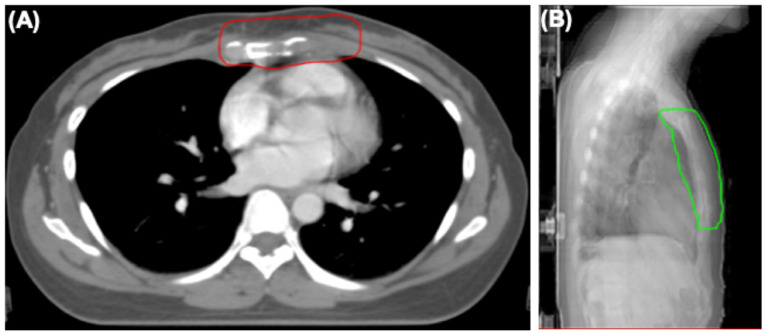
The radiotherapy field of sternum and internal mammary node lesions. (**A**) Axial view showing clinical target (CTV) of sternum and internal mammary node lesions (red line). (**B**) Lateral view showing planning target volume (PTV) of sternum and internal mammary node lesions (green line).

**Figure 5 diagnostics-14-00765-f005:**
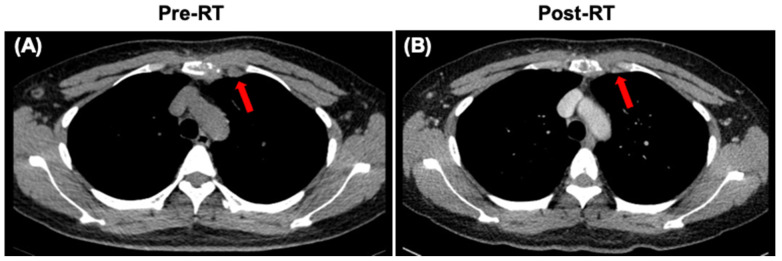
CT images of the sternum and internal mammary node lesions (arrows). (**A**) CT scan showing the sternum and internal mammary node lesions at diagnosis. (**B**) CT scan showing the sternum and internal mammary node lesions after local radiotherapy (RT).

## Data Availability

Not applicable.

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
