# Peer review of "Occult Breast Cancer Presenting as Sternum Pain"

_diagnostics, 2024, doi:10.3390/diagnostics14070765_

Round 1

Reviewer 1 Report

Comments and Suggestions for Authors

This case report describes the diagnosis of breast cancer made through sternal and left internal mammary lymph node metastases without a definite breast primary lesion. The case report is very complete, with all of the information and images one needs to understand the case. It does not, however, add significantly to our understanding of the disease process nor suggest a different approach in workup/management.

Comments on the Quality of English Language

One minor comment is to change "no positive signs" to "no visualized lesions" or "no positive findings" in several places.

Reviewer 2 Report

Comments and Suggestions for Authors

The introduction does not provide sufficient background nor relevant references. The research design was not appropriately elaborated. Used methods should be described. Results need to be presented in line with figures but instead the figures are described without text of the Results. The conclusions need to be highlighted.

Round 2

Reviewer 2 Report

Comments and Suggestions for Authors

The authors corrected the manuscript according to suggestions and properly explained the issues that can not be added.